# The Antiviral Effects of 2-Deoxy-D-glucose (2-DG), a Dual D-Glucose and D-Mannose Mimetic, against SARS-CoV-2 and Other Highly Pathogenic Viruses

**DOI:** 10.3390/molecules27185928

**Published:** 2022-09-12

**Authors:** Beata Pająk, Rafał Zieliński, John Tyler Manning, Stanislava Matejin, Slobodan Paessler, Izabela Fokt, Mark R. Emmett, Waldemar Priebe

**Affiliations:** 1Independent Laboratory of Genetics and Molecular Biology, Military Institute of Hygiene and Epidemiology, Kozielska 4, 01-163 Warsaw, Poland; 2WPD Pharmaceuticals, Zwirki i Wigury 101, 01-163 Warsaw, Poland; 3Department of Experimental Therapeutics, The University of Texas MD Anderson Cancer Center, 1901 East Rd., Houston, TX 77054, USA; 4Department of Pathology, The University of Texas Medical Branch, 301 University Blvd., Galveston, TX 77555, USA; 5Department of Advanced Cardiopulmonary Therapies and Transplantation, The University of Texas Health Science Center at Houston, Houston, TX 77054, USA

**Keywords:** metabolic shift, viral infections, SARS-CoV-2, glycolysis, glycosylation, 2-deoxy-D-glucose, novel analogs

## Abstract

Viral infection almost invariably causes metabolic changes in the infected cell and several types of host cells that respond to the infection. Among metabolic changes, the most prominent is the upregulated glycolysis process as the main pathway of glucose utilization. Glycolysis activation is a common mechanism of cell adaptation to several viral infections, including noroviruses, rhinoviruses, influenza virus, Zika virus, cytomegalovirus, coronaviruses and others. Such metabolic changes provide potential targets for therapeutic approaches that could reduce the impact of infection. Glycolysis inhibitors, especially 2-deoxy-D-glucose (2-DG), have been intensively studied as antiviral agents. However, 2-DG’s poor pharmacokinetic properties limit its wide clinical application. Herein, we discuss the potential of 2-DG and its novel analogs as potent promising antiviral drugs with special emphasis on targeted intracellular processes.

## 1. SARS-CoV-2 Pandemic and Current COVID-19 Treatment

By the end of June 2022, severe acute respiratory syndrome coronavirus 2 (SARS-CoV-2) infection had been diagnosed at 560 M cases causing 6.36 M deaths worldwide [1]. The rapid spread of coronavirus disease 2019 (COVID-19) was declared an emerging public health threat by the World Health Organization. It caused a global pandemic not observed in recent history. The incubation period for COVID-19 was up to 14 days, with a median time of 4–5 days from exposure to symptoms onset [2]. Most people infected with the SARS-CoV-2 virus experienced mild to moderate respiratory illness and recovered without requiring special treatment. However, older people and those with underlying medical problems, such as cardiovascular disease, diabetes, obesity, chronic respiratory disease, and cancer, were more likely to develop severe illness.

Vaccination remains the most effective way to prevent severe illness caused by SARS-CoV-2 infection. However, despite the widespread availability of SARS-CoV-2 vaccines, many individuals are either not fully vaccinated or cannot mount adequate responses to the current vaccines. Some of these people, if infected, are at high risk of progressing to develop more serious COVID-19. The most severe SARS-CoV-2 complication is pneumonia that, according to Grant et al. [3], lasts longer and causes more harm than typical viral pneumonia. While other types of pneumonia rapidly infect large regions of the lungs, COVID-19 begins in numerous small areas of the lungs. As COVID-19 pneumonia slowly moves through the lungs, it leaves damaged lung tissue in its wake. Additionally, fever, low blood pressure, and organ damage have been reported in COVID-19 patients [3].

According to available National Institutes of Health (NIH) treatment guidelines, the standard of care (SOC) for hospitalized COVID-19 patients, with progressive disease development, includes the use of corticosteroids, such as dexamethasone [4], increasing the risk of permitting secondary bacterial and fungal infections, due to the extensive and prolonged immune suppression.

Very recently, the Food and Drug Administration (FDA) approved the additional use of baricitinib (selective inhibitor of Janus kinases 1 and 2 (JAK1/2)) or tocilizumab (a recombinant humanized anti-IL-6 receptor monoclonal antibody) as anti-inflammatory compounds [5] for patients who require a high-flow device for noninvasive ventilation and have rapidly increasing systemic inflammation. On 30 July 2021, the FDA also expanded the Emergency Use Authorization (EUA) indication for the anti-SARS-CoV-2 monoclonal antibodies casirivimab plus imdevimab to allow this combination to be used as post-exposure prophylaxis (PEP). This combination was approved for use in mild to moderate COVID-19 adults and pediatric patients (>12 years old) with a positive result of the SARS-CoV-2 RT-PCR test and at high risk for progression to severe COVID-19, due to the above mentioned underlying medical problems. On 22 December 2021, the FDA granted emergency use authorization for the oral protease inhibitor nirmatrelvir for prevention of severe cases of COVID-19. However, two recent studies indicate that the SARS-CoV-2 virus can mutate its genome making the virus resistant to nirmatrelvir [6,7].

Today, the number of drugs approved for COVID-19 treatment is limited and does not entirely fulfill the medical need for highly efficient, cost-effective, and readily accessible antiviral treatment. Very recently, in May 2021, the Drug Controller General of India gave emergency approval for 2-deoxy-D-glucose (2-DG) in its oral formulation as an adjunct therapy along with the SOC in hospitalized moderate-to-severe COVID-19 patients. According to the phase III clinical trial results, COVID-19 patients treated with 2-DG did not require supplemental oxygen therapy by day three, compared to those treated with SOC (42% vs. 31%), thereby indicating an early recovery of pulmonary function. Notably, a similar trend was observed in patients aged 65 and above [8]. The only information from these trials is available from a government press release and trials’ registrations on the Clinical Trial Registry of India. Both the second and third phases of the 2-DG trials were not double blinded, making interpretation of the results less reliable [9]. Nevertheless, the clinical trials leading to 2-DG approval in India further support the idea of using D-glucose analogs as an antiviral therapy. The rationale for using 2-DG in antiviral treatment is presented below.

## 2. Metabolic Shift in Host Cells during Viral Infection

Viral infections induce virus-specific metabolic reprogramming in host cells [10]. Viral replication entirely relies on the host cell machinery to synthesize viral components, such as nucleic acids, proteins, glycans and lipid membranes [11]. As virus formation depends on the metabolic capacity of the host cell to provide required components and energy in the form of ATP, the majority of viruses modulate the host cell metabolism to optimize the biosynthetic needs for virus growth. Both DNA and RNA viruses have been shown to affect various aspects of host metabolism, including increased glycolysis, elevated pentose phosphate activity, and enhanced amino acid and lipid synthesis. Generally, viruses mainly increase the consumption of key nutrients like glucose and glutamine. However, the precise metabolic changes are often virus-dependent and can vary even within the same family of viruses, as well as with the host cell type [10].

There are multiple ways in which viruses can alter host cell metabolic processes. For example, it was shown that human cytomegalovirus (HCMV), herpesvirus-1 (HSV-1), and adenovirus (ADWT) increase glycolysis and the tricarboxylic acid (TCA) cycle, as well as enhance nucleotide and lipid synthesis [12,13,14].

According to Mayer et al. [15], most virus-infected host cells upregulate glycolysis. It should be explained that under homeostatic and aerobic conditions, cells maintain ATP production mainly by aerobic glycolysis, followed by feeding pyruvate into the TCA cycle and subsequent utilization of reduced molecules into the oxidative phosphorylation pathway. On the contrary, pyruvate is converted to lactate under anaerobic conditions, which is then eliminated to extracellular space. Aside from the anaerobic conditions, Otto Warburg observed that cancer cells utilize glucose mainly via glycolysis, even under normal oxygen conditions (normoxia), the so-called Warburg effect [16]. Cells infected by certain viruses appear to adapt similar metabolic alterations to cope with the high anabolic demands of virion production. Whereas the overall virus-induced metabolic abnormalities are unique and virus-specific, the upregulation of the glycolysis pathway is a common phenomenon, and as such has been considered a target for antiviral therapy. Significantly, increased glycolysis activity has also been described in coronavirus infections, including porcine epidemic diarrhea coronavirus (PED), MERS-CoV, and SARS-CoV-2 [17,18,19,20]. One of the common mechanisms induced by a viral infection that leads to glycolysis is upregulation of the phosphatidylinositol-3-kinase (PI3K)/protein kinase B (PKB/Akt) signaling pathway, which regulates the expression of glucose transporters 1 or 4 (GLUT1, GLUT4) [18,20,21,22]. Elevated expression of GLUTs facilitates an increase in glucose uptake by infected host cells. In response, the level of glycolytic enzymes, such as hexokinase (HK), lactate dehydrogenase (LDHA), or phospho-fructokinase 1 (PFK-1), is also higher [21,23,24]. Viruses that were confirmed to induce the Warburg effect in host cells are summarized in Table 1. Additionally, viruses such as human T cell leukemia virus type 1 (HTLV-1) utilize GLUT1 as a receptor for entry [25]. While there is no literature that investigates the effect of 2-DG on HTLV-1 replication or infection, it is possible that 2-DG could impact HTLV-1 infection through competitive binding with the GLUT1 receptor.

In summary, several epidemiologically significant viruses and their efficient replication in host cells depend on the glycolysis process.

## 3. The Importance of Host Glycosylation Process for Viral Replication

Carbohydrates are one of the most critical components necessary for the synthesis of N-glycans in the endoplasmic reticulum (ER). Numerous viruses rely on the expression of specific viral oligosaccharides crucial for viral entry into the host cells, proteolytic processing, protein trafficking, and evading detection by the host immune system [43,44]. In the *N*-glycosylation process, a high mannose core is attached to the amide nitrogen of asparagine in the context of the conserved motif Asn-X-Ser/Thr. It occurs early in the protein synthesis, followed by a complex process of trimming and remodeling of the oligosaccharide during transit through the ER and Golgi [43]. It has been shown that several viruses hijack the cellular glycosylation pathway to modify viral proteins. Adding N-linked oligosaccharides to the envelope or surface proteins promotes proper folding and subsequent trafficking using host cell chaperones and folding factors. Often, viruses use calnexin and or calreticulin to facilitate the proper folding of overexpressed viral proteins [45]. Although the cell cannot distinguish between host and viral proteins, one difference noted is an increase in the level of glycosylation in many viral glycoproteins. During viral evolution, glycosylation sites are easily added and deleted, increasing the possibility of viral modifications. Glycosylation sites have a significant impact on the survival and transmissibility of the virus and small changes can alter protein folding and conformation, affecting portions of the entire molecule [46]. Further, changes in glycosylation can affect interactions with receptors, influence virus entry, and protect the virus from neutralizing antibodies [44,47]. It was shown that many viruses even use glycosylation for important functions in their pathogenesis and immune evasion, including influenza A and B, HIV, and hepatitis C [23,28,48,49].

The glycosylation process requires mannose, an essential constituent of *N*-glycans. Mannose enters the cells via hexose transporters present in the plasma membrane. It is immediately phosphorylated by HK and then either catabolized via mannose phosphate isomerase (MPI) or diverted toward glycosylation through phoshomannomutase-2 (PMM2) [50]. On the other hand, mannose-6-phosphate (M6P) could also be obtained via the MPI-catalyzed isomerization of fructose-6-phosphate, synthesized from glucose-6-phosphate in the glycolysis pathway [51]. Moreover, a virus-induced metabolic shift in infected host cells results directly in higher activity of HK, which upregulates the glycosylation process that is required for rapid and massive production of infectious progeny in order to disseminate the infection.

Based on the mechanisms mentioned above, inhibition of glycolysis could be a potent antiviral approach. One of the most widely used glycolysis inhibitors is the D-glucose analog, 2-deoxy-D-glucose (2-DG).

## 4. 2-DG in Antiviral Research

### 4.1. 2-DG Molecule and Its Intracellular Effects

The D-glucose analog, 2-DG has been the primary compound for glycolysis inhibition for a long period of time. 2-DG is a synthetic analog of glucose in which hydrogen replaces the hydroxyl group at the second carbon position (Figure 1A). Similar to D-glucose, 2-DG is taken up by the cells, mainly via glucose transporters (facilitated diffusion), in particular, GLUT1 and GLUT4, although active transport SGLT transporters also occurs [52,53]. Once intracellular, 2-DG is phosphorylated to 2-deoxy-D-glucose-6-phosphate (2-DG-6-P), producing a charged compound now entrapped inside the cell. However, because it is missing the 2-OH group it cannot undergo isomerization to fructose-6-P, therefore, it accumulates in the cell and causes inhibition of glycolysis and glucose metabolism [54] (summarized in Figure 1B). 2-DG inhibits hexokinase and phospho-hexose isomerase responsible for conversion of phosphoglucose to phosphofructose and, thereby, blocks glycolysis at the initiation stage. Glycolysis inhibition results in depletion of ATP required for maintaining intracellular processes and, thus, facilitates autophagy and apoptosis initiation. Insufficient ATP levels inside the host cells also limit the possibility of fast viral replication and new virus production, thus limiting viral infection.

Historically, 2-DG was synthesized from D-glucose by the elimination of the hydroxyl group at C-2. However, eliminating the hydroxyl group at C-2 in the D-mannose molecule leads to the same 2-DG compound (Figure 1A). Thus, 2-DG can interfere with the metabolism of both D-glucose and D-mannose. Disrupting mannose-related metabolic pathways leads to dysregulation of the *N*-glycosylation process, a crucial cellular process required for production of viral glycoproteins and virions [55]. As 2-DG is substituted into the growing N-glycan chain for mannose, the oligosaccharide chain is truncated, due to the lack of a hydroxyl group in the C-2 position incorporated in the place of mannose (Figure 1C). The 2-DG-dependent inhibition activity of HK also affects mannose processing to further increase the negative effect on production of viral glycoproteins.

As an analog of D-mannose, 2-DG diminishes the cellular pool of mannose for protein glycosylation. Insufficient protein maturation results in lower quality of viral glycoproteins and induces ER stress called “unfolded protein response (UPR)”. The UPR completely shuts down further protein synthesis to alleviate this stress. Consequently, protein synthesis becomes limited and viral envelope formation is disrupted, resulting in inhibition of viral infection.

Inhibition of glycolysis results in limited ATP generation, which is essential for maintaining cellular function and molecular synthesis. In response to the low AMP/ATP ratio, the autophagy process is induced [56]. Prolonged autophagy and/or UPR stress are known to be apoptosis inducers leading to cell death [57,58]. Moreover, 2-DG action has been reported to generate ROS, which is harmful for proteins, nucleic acids, and other intracellular molecules, facilitating programmed cell death [59]. Our team has previously published a detailed description of the intracellular effects of 2-DG action [54].

### 4.2. Antiviral Action of 2-DG

Due to the above-mentioned biological properties of 2-DG and its ability to interfere with various cellular processes, 2-DG is an efficient cytotoxic agent that was tested in different models of viral infections. 2-DG has been explored as a single antiviral agent or as an adjuvant agent for various groups of clinically used drugs.

#### 4.2.1. SARS-CoV-2 and Other Coronaviruses

As mentioned before, 2-DG is a drug candidate for SARS-CoV-2, and its emergency approval in India allowed its clinical use in COVID-19 patients. Numerous in vitro studies have shown that 2-DG efficiently limits SARS-CoV-2 replication [60,61,62]. According to Bhatt et al. [61], SARS-CoV-2 infection of Vero E6 cells induced upregulation of GLUT1, GLUT3, and GLUT4 proteins. GLUT3 is the main transporter determining the high influx of glucose into the cell confirmed by using a fluorescent 2-DG analog (2-NBDG). In addition, increased levels of key glycolytic enzymes, such as HKII, PFK-1, and pyruvate kinase 2 (PKM-2), were also observed [61]. Importantly, 2-DG did not exert a cytotoxic effect on non-infected cells up to 5 mM concentration. In SARS-CoV-2-infected cells, 2-DG [5 mM] reduced cytopathic effects and cell death. Moreover, 2-DG was shown to disrupt the glycosylation of viral proteins, leading to reduced infectivity observed by newly formed virions from collected media [62]. Results demonstrated by Bhatt et al. [61] agree with data published by Bojkova et al. [62], showing the inhibitory effect of 2-DG against SARS-CoV-2 replication in the Caco-2 cell line. The IC_50_ value was estimated to be 9.09 mM [62]. Further, Codo et al. showed that a metabolic shift also occurred in inflammatory cells, such as monocytes, in response to SARS-CoV-2 infection [63]. Due to impaired oxidative metabolism, HIF-1α protein becomes upregulated in infected monocytes, resulting in a prolonged pro-inflammatory state. This, in turn, leads to pro-inflammatory cytokine production, which further deteriorates neighboring cells in a paracrine way, including T-cells [63].

Before the onset of the SARS-CoV-2 pandemic, 2-DG was already recognized as an efficient antiviral compound against other coronaviruses. In 2014, Wang et al. [17] examined the influence of 2-DG on the porcine epidemic diarrhea virus (PEDV). The authors found that 2-DG [10 mM] inhibits PEDV replication in Vero cells, mainly affecting the glycosylation process and UPR stress induction [17].

Altogether the above data demonstrate that inhibition of glycolysis/glycosylation is an effective strategy to limit coronavirus infection. Thus 2-DG’s implication as a clinical anti-viral therapy is promising and well justified.

#### 4.2.2. Papillomaviruses

2-DG [7.5 mg/mL] was shown to be capable of suppressing the transcription of the human pathogenic papillomavirus type 18 (HPV18) in HeLa cells [64]. Interestingly, the authors found that using the intracellular Ca^2+^ antagonist–TMB-8, the 2-DG effect can be abolished. However, the mechanism connecting 2-DG and calcium signaling has not been explained. Antiviral 2-DG action against HPV18 was also demonstrated by Kang et al. [65] who showed that 2-DG downregulates the Sp1 transcription factor activity, leading to restricted HPV18 early gene expression. The molecular mechanism of 2-DG action preferably affected the glycolysis process, whereas the reduced ATP generation was involved only to a limited extent [65].

The importance of HPV-16-mediated metabolic shift was also demonstrated in studies by Ma et al. [66]. It has been reported that E6 and E7 HPV oncoproteins, which contribute to viral-induced cervical carcinogenesis, also determine cervical cancer resistance to 5-fluorouracil (5-FU) treatment, due to the upregulated glycolysis and Akt-dependent signaling pathway [66]. In the presence of 2-DG [1 mM], virus-induced glycolysis was inhibited, and cervical cancer was sensitized to 5-FU cytotoxic action.

It should be emphasized that with around 604,000 cases and 341,000 deaths in 2020 [67], cervical cancer is the fourth most common cancer worldwide and the most common malignant transformation caused by HPV infection. It is estimated that about 90% of all women contract an HPV infection in their lives. In about 10% of cases, the virus persists, and cervical intraepithelial neoplasia (CIN) develops. Approximately 1% of women with high-risk HPV infection will develop a cervical carcinoma within 1 to 20 years [68]. The efficient inhibition of HPV replication using 2-DG is an exciting approach that should be verified in the clinics.

#### 4.2.3. Rhinoviruses

Rhinoviruses (RV) are the causative agents of the common cold and other respiratory tract infections. Despite the vast prevalence, effective treatment or prevention strategies are still lacking [69]. Previously, it was shown that RV infection also induces a metabolic shift in infected host cells making it potentially susceptible to antiviral effects of 2-DG. According to Gualdoni et al. [39], 2-DG administration [5 mM] to RV-infected HeLa cells reversed many RV-induced modifications in cellular metabolism. For instance, 2-DG abolished RV-induced glycogenolysis and led to a significant increase in the levels of several fatty acylcarnitine’s that were decreased during infection. These changes were accompanied by reduced levels of various phospholipids, sphingolipids, and ceramides, which, taken together, suggest a shift away from anabolic and lipogenic processes during 2-DG treatment [39]. In vivo analysis of RV infection in the murine model showed reduced lung inflammation in 2-DG-treated animals [5 mM] with no visible side effects upon treatment. Therefore, 2-DG might be considered as a strategy to combat this widespread pathogen.

#### 4.2.4. Noroviruses

Noroviruses (NoV) are nonenveloped, positive-sense, single-stranded RNA viruses of the *Caliciviridae* family that cause acute, nonbacterial gastroenteritis globally. Murine norovirus (MNV) infection of macrophages causes changes in the host cell metabolic profile characterized by an increase in central carbon metabolism. Energetic profiling, combined with experiments inhibiting the pentose phosphate pathway (PPP) and OXPHOS with 6-aminonicotinamide and oligomycin A, respectively, revealed that these pathways have a minor role in murine MNV pathogenesis compared to glycolysis. Investigations of Akt and AMPK pathways showed that MNV infection caused an increase in Akt activation, while inhibition of Akt signaling reduced both cellular glycolysis and MNV infection [20]. Downregulation of glycolysis with 2-DG [10 mM] treatment significantly reduced MNV infection in RAW 264.7 cell line. In contrast, 2-DG was ineffective against human astrovirus in vitro, suggesting that metabolic changes and viral dependence upon selected intracellular processes might be virus-specific [20].

#### 4.2.5. Hepatitis B Virus

Hepatitis B virus (HBV) is a partially double-stranded circular DNA virus whose genome is approximately 3200 bases with four overlapping open reading frames (ORFs) and it belongs to *the Hepadnaviridae* family. HBV prevalence varies worldwide, with high rates reported in low-income countries. Approximately 90% of HBV infections are acute, while 10% progress to chronic infection among adult patients [70]. As an intracellular pathogen, the reproduction of HBV depends on the occupancy of host metabolism. Wu et al. [29] showed that large viral surface antigens (LHBS) interact directly with cellular PKM-2, a key regulator of glucose metabolism in hepatocytes, thereby increasing glucose utilization and lactate production. Next, the authors showed that 2-DG treatment [0.5–10 mM] caused dose-dependent suppression of HBV protein synthesis, leading to inhibition of viral replication [29]. Supporting data showing the potency of 2-DG action in HBV treatment have been published by Wang et al. [30]. 2-DG treatment [1, 5, 10 mM] significantly decreased glycolysis in HepG2.2.15 cells with concomitant reduction of intra- and extracellular HBV DNA and RNAs, and the addition of pyruvate did not affect 2-DG action. Moreover, 2-DG modulated the cellular AMP/ATP ratio, thereby activating AMPK kinase and autophagy. These data confirmed that 2-DG could inhibit glycolysis, HBV gene expression, and replication in HepG2.2.15 cells.

Strikingly, chronic hepatitis B infection affects more than 300 million people worldwide and is a leading cause of liver failure and cancer [71]. Although current treatments for chronic HBV suppress viral replication and reduce the risk of liver cancer and end-stage liver disease, it does not constitute a complete virus elimination. Thus, treatment interruption may result in a resurgence of viral replication and hepatic disease progression.

Targeting infected host cell metabolism via 2-DG or other glycolysis inhibitors may represent a viable approach that needs to be clinically tested.

#### 4.2.6. Zika Virus

Zika virus (ZIKV), a mosquito-transmitted flavivirus, spread in recent years from Africa and Asia to Latin America and parts of the United States. It has rapidly emerged as an important pathogen that can cause significant morbidity [42]. Singh et al. [42] showed that ZIKV requires inhibition of AMPK signaling and concomitant upregulation of glycolysis to promote viral replication. This corresponds to increased glucose uptake and mRNA expression of GLUT1, HK2, triosephosphate isomerase (TPI), and monocarboxylate transporter 4 (MCT4) in infected HReEC endothelial cells. The glycolysis induction is a crucial mechanism for successful ZIKV infection. The use of 2-DG [1 mM] markedly reduced the number of ZIKV Ag-positive HRvEC cells and viral titer relative to untreated cells. Further, 2-DG treatment increased the phosphorylation of AMPK and restored its activation upon ZIKV challenge. Moreover, ZIKV NS3 protein expression was undetectable during 2-DG treatment [42]. In studies performed by Lin et al. [72], 2-DG [10 mM] was also confirmed to inhibit ZIKV replication in the Vero cell culture model.

ZIKV infection during pregnancy can cause microcephaly in newborns, yet the underlying mechanisms remain largely unexplored. Very recently, Pang et al. [73] showed that ZIKV infection caused aberrant metabolism in infected brains. Using LC-MS global proteomic data, the authors were able to identify the enriched pathways in ZIKV-infected brains related to amino acid, purine and pyrimidine metabolism. Downregulated pathways included the TCA cycle, OXPHOS, and pyruvate metabolism [73]. The observed inhibition of the OHPHOS and TCA cycle occurred in neurons and neuroblast cells, suggesting a correlation between mitochondrial dysfunction and ZIKV-induced neural cell death. Additionally, downregulated purine and pyrimidine metabolism toward RNA and DNA synthesis at the protein level implied a low proliferation state for cells of ZIKV-infected mouse brains. Limited glucose utilization via TCA and OHPHOS promote a glycolytic shift in infected neurons that could be targeted with 2-DG treatment. Taken together, these results confirmed the importance of glycolysis for ZIKV replication and showed the potential of 2-DG treatment in ZIKV infection.

#### 4.2.7. Herpes Simplex Virus 1

The herpes simplex virus (HSV) is the causative agent of herpes infection. Herpes can appear on various parts of the body, most commonly on the genitals and mouth. There are two types of HSV: HSV-1–primarily causes oral herpes and is generally responsible for cold sores and fever blisters around the mouth and on the face; HSV-2–primarily causes genital herpes and is generally responsible for genital herpes outbreaks [74]. According to the WHO, about 3.7 billion people under age 50 (67%) have HSV-1 infection, whereas 492 million people aged 15–49 (13%) worldwide have HSV-2 infection [75].

Abrantes et al. [24] showed that HSV-1 induces glycolysis in infected cells via upregulation of PFK-1 activity leading to increased ATP content inside the host cells. Interestingly, no data confirm a similar metabolic shift in HSV-2-infected cells. According to Varanasi et al. [76], 2-DG action against HSV-1 changes during different stages of HSV pathogenesis and can have either detrimental or beneficial effects. In the case of HSV-1 infection in mice, upregulated metabolism and glucose uptake was observed in CD4 T cells compared with T cells from naive animals. Treatment with 2-DG reduced glucose uptake and limited the differentiation of effector T cells in the in vitro model. On the other hand, in vivo results demonstrated that 2-DG treatment diminished SK lesions, due to reduced effector T cell responses. In this context, 2-DG appeared to inhibit HSV-1 infection via modulating inflammatory CD4 effector T cells response, resulting in damaging consequences in the unique environment of the eye [76]. On the contrary, 2-DG administration in the acute phase of ocular infection resulted in death from herpes encephalitis in many animals. Taken together, Varanasi et al. [76] concluded that metabolic modifying drugs should be used with caution, especially during HSV-1 infections. When 2-DG therapy was used when the HSV virus was still replicating, viral replication was enhanced, which could have had lethal consequences due to the virus spreading to the brain.

Distinct observations concerning 2-DG effects against HSV-1 infection have also been reported by Knowles and Person [77]. 2-DG [10 mM] and glucosamine were found to inhibit cell fusion caused by a syncytial mutant of HSV and to also inhibit glycosylation of viral glycoproteins in infected HEL cells. These effects were substantially reduced when mannose was also present during infection. The correlation between fusion and glycosylation in the presence of 2-DG and mannose suggests that the cells cannot fuse if their glycoproteins have a considerably reduced carbohydrate content [77]. According to the presented data, 2-DG appeared to affect the glycosylation process that was crucial for cell fusion. However, the authors did not evaluate the glycolysis process in HEL cells, and the 2-DG effect on glycolysis was not discussed.

On the other hand, studies published by Kern et al. [78] and Shannon et al. [79] demonstrated a lack of antiviral action of 2-DG in the treatment of cutaneous infections with HSV-1 in mice and genital infections with HSV-2 in mice [78] and guinea pigs [78,79]. In all experimental models, 2-DG treatment (topically (HSV-1)/intravaginally (HSV-2), three times a day with 0.2% or 0.5% 2-DG solution beginning 3 h after inoculation) did not significantly affect viral replication, lesions development, severity, mortality, or latency.

In summary, studies testing 2-DG efficacy in HSV infections are limited, and there is no significant progress in this area. It seems that positive cell culture data does not translate into positive outcomes in animal studies. We hypothesize that a lack of satisfactory in vivo 2-DG effects could be correlated to the poor pharmacokinetic properties of 2-DG [54]. That issue has been discussed previously and is supported by clinical data from 2-DG studies in the past.

## 5. 2-DG in Clinical Trials

Due to 2-DG’s ability to inhibit glycolysis, ATP synthesis and protein glycosylation, 2-DG appears to be very efficient in killing highly glycolytic cells. As mentioned previously, metabolic shift is characteristic of viral infection and cancer cells. Importantly, all described 2-DG effects are mostly observed in glycolytic cells, without significant influence on the viability of normal cells [80]. Thus, 2-DG has been explored as a cytotoxic compound or an adjuvant agent for various clinically used chemotherapeutic drugs in breast, prostate, ovarian, lung, glioma, and other cancer types. 2-DG was also tested as a radio-sensitizing agent in cancer radiotherapy. The efficacy of 2-DG as an anticancer agent was reviewed in detail in our paper [54]. Due to the importance of cancer treatment for global population healthcare, 2-DG has been tested in oncological clinical trials. Clinical trials registered in India using 2-DG in COVID-19 patients are the first documented cases for clinical use of 2-DG in viral infections.

Despite the numerous preclinical and clinical studies, the use of 2-DG in cancer and viral treatment has been limited. Its rapid metabolism and short half-life (according to Hansen et al., after treatment with infusion of 50 mg/kg^2^-DG, its plasma half-life was only 48 min [81]), make 2-DG a relatively poor drug candidate. Moreover, 2-DG must be given at relatively high concentrations (≥5 mmol/L) to compete with blood glucose [82]. According to Stein et al. [83], the dose of 45 mg/kg received orally on days 1–14 was defined as safe because patients did not experience any dose-limiting toxicities. Notably, at the dose of 60 mg/kg, two patients experienced dose-limiting toxicity of grade 3–asymptomatic QTc prolongation. According to former studies published by Burckhardt et al. [84] and Stalder et al. [85], among patients exposed to 2-DG, non-specific T wave flattening and QT prolongation, without any event of severe arrhythmia, developed.

A study of 2-DG in humans was published in 2013 and reported the results of an association regimen of 2-DG and docetaxel in patients with advanced solid tumors [86]. In this study, based on the overall tolerability of the 2-DG treatment, the authors used a starting dosage of 63 mg/kg, which was considered safe. At the higher dose of 88 mg/kg, patients presented plasma glucose levels above 300 mg/dL and glucopenia symptoms, including sweating, dizziness, and nausea, mimicking the symptoms of hypoglycemia [86]. Other significant adverse effects recorded during the trial at 63–88 mg/kg doses were gastrointestinal bleeding (6%) and reversible grade 3 QTc prolongation (22%). After the end of the study, one patient died from a serious adverse event of cardiac arrest 17 days after the last dose of 2-DG. ECG done ten days before death showed persistent T-wave inversion and no QT prolongation [86]. However, it should be noted that the eligibility criteria of patients in this study, who had advanced or metastatic solid tumors, could have played a confounding role in relation to survival and overall patient condition. Clinical testing of 2-DG as a chemotherapy has been performed in humans and demonstrated good tolerability. Antiviral efficacy of 2-DG has been demonstrated in various models and showed a good tolerability profile. Currently, there are no available reports presenting data about safety and efficacy of 2-DG in the COVID-19 clinical trials. It is also possible, if not likely, that some hospitalized patients receiving i.v. fluids may also receive glucose at 5%, 10% or even higher (not limited to COVID-19 patients). This can be especially the case for patients that are intubated and unable to drink and eat. However, our primary goal is to reduce hospitalization rates and improve recovery through early treatment of patients that are not receiving i.v. fluid therapy yet, especially in outpatient groups of COVID-19 infected population.

Based on the available data, our group is not aware of any specific negative impact of this type of therapy against other viral infections. The only exception could be related to Herpes infection and that is being addressed in the other section of the paper. In general, it is certainly possible that patients receiving glucose i.v. therapies may not benefit from 2-DG, but we can only speculate at this stage.

Nevertheless, the above-described poor pharmacokinetic properties and possible side effects encourage identification of other molecules that affect the same metabolic pathways but could overcome these problems. One possible solution is the use of novel 2-DG analogs, which maintain 2-DG-mediated biological efficacy, but have better drug-like properties, which is essential for successful clinical introduction.

## 6. Novel 2-DG Analogs and Their Potential for Antiviral Therapy

Acetyl-2-DG analogs has been developed in Dr. Waldemar Priebe’s laboratory. Among tested derivatives, lead compound WP1122 (3,6-di-O-acetyl-2-deoxy-D-glucose) has been selected as a potent glycolysis inhibitor (Figure 2).

Compound WP1122 was prepared from commercially available 3,4,6-tri-O-acetyl-D-glucal in a two steps synthesis. At the first step 3,4,6-tri-O-acetyl-D-glucal was selectively deacetylated to 3,6-di-O-acetyl-D-glucal, which, in the next step, was treated with water solution of hydrobromic acid to give WP1122 as the final product.

WP1122 enters the cells via passive diffusion rather than relying upon specific glucose transporters. Inside the cells, WP1122 undergoes deacetylation by esterases releasing active 2-DG molecules (Figure 2). Further, 2-DG undergoes phosphorylation at the C-6-hydroxyl group, and it is trapped inside the cells. 6-phosho-2-DG competitively inhibits HK, blocking phosphorylation of glucose and thereby inhibiting the glycolytic pathway [86]. Furthermore, it has been shown that WP1122 crosses the blood-brain barrier (BBB), making it a promising drug candidate for glioma therapy [87] and possibly viral encephalitis. 2-DG is rapidly metabolized, whereas the prodrug WP1122 releases 2-DG slowly, increasing its half-life. WP1122 demonstrated good oral bioavailability, resulting in a two-fold higher plasma concentration of 2-DG than that achieved via administration of 2-DG alone [87]. In vitro studies showed that WP1122 effectively inhibits glycolysis with 2–10 times more potent action when compared to 2-DG. Moreover, WP1122 was well tolerated by mice in an orthotopic glioma model, even with prolonged exposure [88]. WP1122 is currently licensed to Moleculin Inc. and is in phase 1 clinical trials in COVID-19 patients. According to a statement by Moleculin Inc. [89], WP1122 antiviral action has been tested in cooperation with Goethe University in Frankfurt in Germany and showed complete inhibition of SARS-CoV-2 replication in cell culture. The data indicated that WP1122 could be more beneficial clinically than 2-DG alone.

The other group of 2-DG analogs are halogenated D-glucose derivatives, described previously by Lampidis et al. [90], such as 2-fluoro-2-deoxy-D-glucose (2-FG), 2-chloro-2-deoxy-D-glucose, 2-chloro-2-deoxy-d-glucose (2-CG), and 2-bromo-2-deoxy-D-glucose (2-BG) [90], see Figure 3.

The authors also evaluated the ability of halo-derivatives to interact with HKI enzyme and their cytotoxic potential against glycolytic cancer cells [90]. It appeared that there was a negative correlation between the size of halogen substituent at the C-2 position and drug activity. As halogen size increased (2-FG > 2-CG > 2-BG), the ability to bind the HKI active site reduced, leading to diminished production of 6-O-phosphorylated intermediates, which is crucial for glycolysis inhibition. Interestingly, the authors did not analyze the iodo-analogs that, according to a recent analysis published by Ziemniak et al. [91], could also have an inhibitory potential against HK activity. Further studies are needed to verify whether halo-analogs could also exert antiviral effects in infected cells.

## 7. Perspectives

The SARS-CoV-2 pandemic reminded societies globally of the importance of viral diseases in human health. The rapid spread of SARS-CoV-2 and its millions of infected patients have demonstrated the lack of effective broad-spectrum antiviral treatments. Moreover, as described above, other viral infections like HBV, HPV, HSV and ZIKV also have significant health, economic and worldwide significance. All of them generate a high demand for an effective therapy that reduces infections and protects patients from the long-term harmful consequences of viral diseases, including cancer patients. Activation of glycolysis in infected cells is the common link between various viral infections making inhibition of glycolysis a promising therapeutic approach for broad-spectrum drugs. As can be seen from the numerous studies described above, 2-DG exhibits effective antiviral activity against many types of viruses, including SARS-CoV-2. Recent clinical trials of 2-DG in SARS-CoV-2-infected patients support the strategy of targeting the metabolism of infected host cells as a way to limit virus growth and dissemination in infected host. However, based on the cited data on the effects of 2-DG clinical trials for oncological indications, including the reported side effects and poor pharmacokinetic properties, it seems that there is an unmet need to search for new molecules with an analogous mechanism of action but with significantly better drug-like properties.

In this light, molecules like WP1122 appear to have great potential for development as a drug candidate in antiviral indications. We look forward to the final reports of clinical trials with WP1122 in patients with COVID-19 or other viral infections of public health importance.

## Figures and Tables

**Figure 1 molecules-27-05928-f001:**
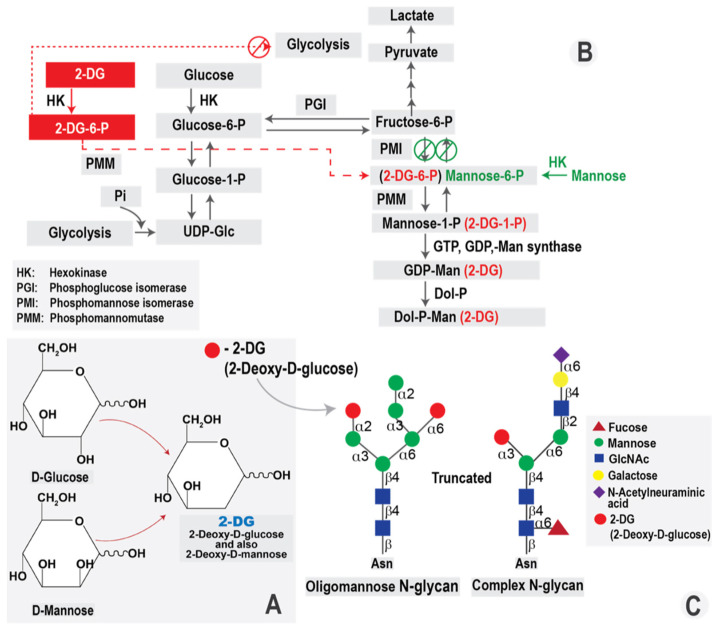
2-Deoxy-D-glucose (2-DG) and its biological effects. (**A**). Structures of D-Glucose, D-Mannose and 2-DG; (**B**). The effects of 2-DG on the glycolysis pathway and substitution for mannose in N-glycan synthesis; (**C**). Truncation of *N*-glycosylation by 2-DG.

**Figure 2 molecules-27-05928-f002:**
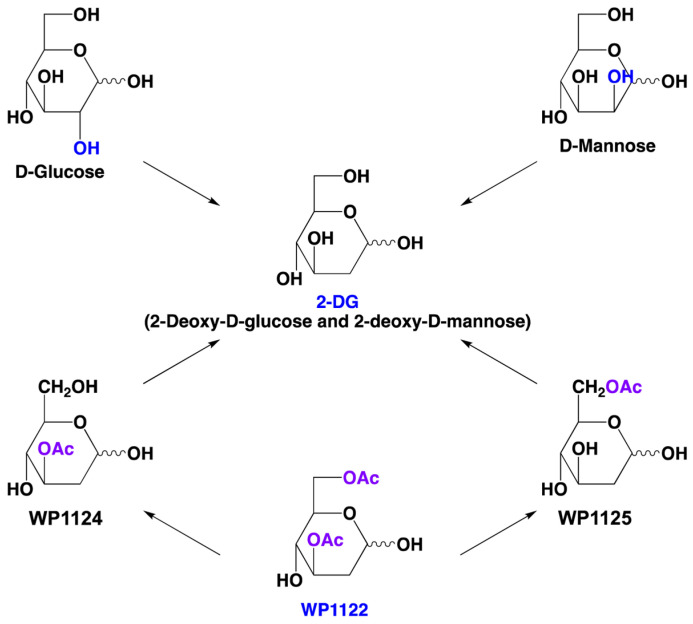
Structures of D-mannose, D-glucose, 2-DG (2-deoxy-D-glucose; 2-deoxy-D-mannose), and WP1122 (2-DG prodrug). Within the cell, WP1122 is deacetylated by cellular esterases to monoacetates: WP1124 and WP1125, and subsequently release active 2-DG.

**Figure 3 molecules-27-05928-f003:**
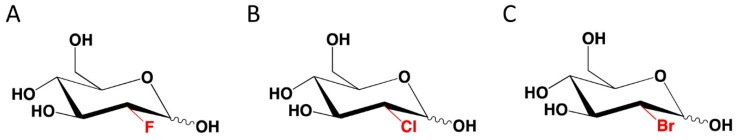
Structures of (**A**). 2-deoxy-2-fluoro-D-glucose (2-FG), (**B**). 2-deoxy-2-chloro-D-glucose (2-CG), and (**C**). 2-deoxy-2-bromo-D-glucose (2-BG).

**Table 1 molecules-27-05928-t001:** Viruses upregulating host cell glycolysis as a mechanism required for their optimal replication.

Virus	Mechanism of Glycolysis Upregulation	References
Murine Norovirus (MNV)	Upregulation of Akt signaling that stimulates glycolysis and glucose metabolism	[20]
SARS-CoV-2	Upregulation of PI3K/Akt signaling and GLUT1 expression	[18,26]
MERS-CoV	PI3K/Akt and MAPK/ERK signaling pathways upregulation	[27]
Porcine epidemic diarrhea virus (PEDV)	Not specified	[17]
Dengue virus	GLUT1 and HK upregulation	[21]
Hepatitis C virus (HCV)	Upregulated HK activity via direct interaction with viral NS5A protein; downregulation of mitochondrial activity, and HIF-1 level upregulation	[23,28]
Hepatitis B virus (HBV)	Upregulated PI3K/Akt/mTOR pathway and GLUT1 expression; interaction with pyruvate kinase isoform M2	[29,30,31]
Human immunodeficiency virus (HIV)	Upregulated GLUT1 expression	[32,33]
Herpes simplex type 1 virus (HSV-1)	Upregulated PFK-1 activity	[24]
Human cytomegalovirus (HCMV)	Upregulation of GLUT4 expression	[22]
Mayaro virus	Enhanced PFK-1 activity and fructose 2,6-biphosphate level	[34]
Infectious spleen and kidney necrosis virus (ISKNV)	Upregulation of glycolytic enzymes expression	[35]
Human T cell leukemia virus (HTLV)	GLUT1 transporter-mediated virus entry	[25]
Human adenovirus type 2 (Ad2)	Viral oncoprotein E4ORF6 upregulates glycolysis pathway proteins expression	[36,37]
Kaposi’s sarcoma-associated herpesvirus (KSHV)	Viral protein ORF45 regulates transcription of glycolysis proteins	[38]
Rhinovirus (RV)	Activation of PI3K/Akt pathway and GLUT 1 expression	[39]
Influenza A virus (AIV)	Increased glucose uptake, glycolytic enzymes activity, and lactate synthesis; detailed mechanism not described	[40]
Human respiratory syncytial virus (HRSV)	Diminished TCA activity and upregulated glycolysis	[41]
Zika virus (ZIKV)	Downregulation of AMPK, upregulation of GLUT1, HK, and other glycolytic genes	[42]

## Data Availability

Not applicable.

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
