# Peer review of "The Antiviral Effects of 2-Deoxy-D-glucose (2-DG), a Dual D-Glucose and D-Mannose Mimetic, against SARS-CoV-2 and Other Highly Pathogenic Viruses"

_molecules, 2022, doi:10.3390/molecules27185928_

Round 1

Reviewer 1 Report

In this manuscript, PajÄ…k et al. summarize the effects of 2-DG on various viruses, including SARS-CoV-2. They also show that WP1122, an analog of 2-DG, is more effective than 2-DG in inhibiting viral replication. 

This review is interesting and presents the effects of 2-DG on each virus, however the validation with the data presented is not sufficient. Also, for practical use of the molecules, the biological effects of 2-DG and WP1122 need to be fully investigated. My specific comments are listed below. 

 Major issues

  1. The authors state that administration of 2-DG does not affect normal cells, but it is necessary to indicate the effect of these molecules in more detail especially in immune cells. 
  2. In connection with the above, the authors shold indicate why the effective dose varies with the virus species but does not affect normal cells. The manuscript states that for coronaviruses, the concentration that does not affect normal cells is 5 mM, while for papillomaviruses, 7.5 mM is effective on infected cells. 
  3. The manuscript states that 2-DG inhibits the glycolytic system. The authors should clearly describe the cytotoxicity in organs such as heart and liver, that utilize much of the glycolytic system.  
  4. Although most of the experiments are conducted using cell lines, it would be better to prove the effects of 2-DG in in vivo experiments using mice and other animal models. 
  5. Please make clear whether 2-DG could inhibit the virus replication? In this case, what part of the viral life cycle could be affected? The authors claim that 2-DG inhibits glycosylation of viral spike proteins, but non-envelope viruses including norovirus could be suppressed by 2-DG. 

Minor issues 

  1. In Table 1., the lines are narrow and difficult to read, so the authors should draw a dotted line for each virus species. 
  2. For HTLV1, GLUT1 is an infection receptor, does 2-DG affect the infection of this virus more specifically? 
  3. In Figure 3, it would be easier to understand if the metabolic pathway of WP1122 is described. 

Author Response

This review is interesting and presents the effects of 2-DG on each virus, however the validation with the data presented is not sufficient. Also, for practical use of the molecules, the biological effects of 2-DG and WP1122 need to be fully investigated. My specific comments are listed below. 

  • Thank you for comments. We corrected our manuscript according to Your suggestions. Please, find below our responses to Your comments.

Major issues

  1. The authors state that administration of 2-DG does not affect normal cells, but it is necessary to indicate the effect of these molecules in more detail especially in immune cells. 
  • 2-DG has been tested in numerous studies concerning cancer and viral-infected cells. The common link between these two models is the upregulation of the glycolysis process, which is the major source of glucose utilization. 2-DG competes with glucose for GLUTs transporters to enter the cells. Since 2-DG is a glucose analog, it affects all cells in the organism. However, due to the rapid glucose uptake in cancer and viral-infected cells, 2-DG accumulation in these cells is significantly higher and allows 2-DG action in lower concentrations than in normal cells. This property has already been widely used in PET imaging, where radiolabeled F-DG is used. There are several papers regarding 2-DG and its mechanism of action in various models, including toxicology data from clinical trials (also discussed in the review). Our goal was to describe current knowledge about 2-DG and its analogs as antiviral drugs. Thus, we did not discuss in detail 2-DG action in other cellular models. We hope that recent clinical trials with 2-DG in India and WP1122 (phase 1) in England will bring additional knowledge concerning 2-DG and WP1122 safety. 

  1. In connection with the above, the authors should indicate why the effective dose varies with the virus species but does not affect normal cells. The manuscript states that for coronaviruses, the concentration that does not affect normal cells is 5 mM, while for papillomaviruses, 7.5 mM is effective on infected cells. 

- Coronavirus experiments were performed in Vero E6 cells, whereas the papillomavirus studies were performed in HeLa cells (African green monkey kidney vs. human cervical carcinoma cells). Any differences in cytotoxicity are likely attributed to differences between the cell lines. The 2-DG concentration in HeLa cells for the papillomavirus experiments should be listed as 7.5 mg/mL instead of 7.5 mM, and has been corrected. This concentration is simply the dose given to observe the effect. The authors make no mention of cytotoxic effect in the paper.

  1. The manuscript states that 2-DG inhibits the glycolytic system. The authors should clearly describe the cytotoxicity in organs such as heart and liver, that utilize much of the glycolytic system.  

- As mentioned above, 2-DG has already been studied in several clinical trials. Some of them are also summarized in the review and describe the 2-DG clinical safety: line 492-513. No severe side effects were observed in the liver in any of the cited reports. With regard to cardiac safety, as mentioned in the text, the use of 2-DG concentration at the dose of 60 mg/kg, two patients experienced dose-limiting toxicity of grade 3 – asymptomatic QTc prolongation. Up to dose of 45 mg/kg received orally on days 1-14 was defined as safe because patients did not experience any dose-limiting toxicities.  It should also be noteworthy that 2-DG has been tested in cancer-related clinical trials. There are no available data describing 2-DG safety in long-term administration in virus-infected patients. As mentioned above, we hope the ongoing clinical trials with 2-DG in India and WP1122 in England will bring additional data concerning 2-DG safety in patients.

  1. Although most of the experiments are conducted using cell lines, it would be better to prove the effects of 2-DG in in vivo experiments using mice and other animal models. 

- The reviewer is correct that in vivo data would be better than cell culture data and that is the direction that we are moving toward. The reviewer is missing the point that the manuscript is a Review of the current state of 2-DG as an anti-viral.  We hope the ongoing clinical trial with 2-DG in India will bring more data concerning the 2-DG efficacy in anti-viral action against at least SARS-CoV-2. We believe that introducing 2-DG into anti-SARS-CoV-2 human therapy will encourage other groups to do 2-DG or its analogs' antiviral research in animal models.

  1. Please make clear whether 2-DG could inhibit the virus replication? In this case, what part of the viral life cycle could be affected? The authors claim that 2-DG inhibits glycosylation of viral spike proteins, but non-envelope viruses including norovirus could be suppressed by 2-DG. 
  • As mentioned in the text, line 103: “According to Mayer et al. [15], most virus-infected host cells upregulate glycolysis. It should be explained that under homeostatic and aerobic conditions, cells maintain ATP production mainly by aerobic glycolysis, followed by feeding pyruvate into the TCA cycle and subsequent utilization of reduced molecules into the oxidative phosphorylation pathway. On the contrary, pyruvate is converted to lactate under anaerobic conditions, which is then eliminated to extracellular space”. And line 93: “Both DNA and RNA viruses have been shown to affect various aspects of host metabolism, including increased glycolysis, elevated pentose phosphate activity, and enhanced amino acid and lipid synthesis”. Upregulated glycolysis is a key process for rapid viral replication using host machinery both envelope and non-envelope viruses. Its inhibition limits nucleic acid replication and new virus formation within the infected cells for both enveloped and non-enveloped viruses. We gave many examples of inhibition of viral replication/infectivity is section 4.2.

The reviewer mentions noroviruses, which are actually glycosylated despite being non-enveloped (Hanisch, Glycobiology 2002, 32(6), 496-505).  The review does focus on glycolysis inhibition as the mechanism that impacts virus replication in this case.  Furthermore, glycolysis process is connected with mannose synthesis, that is crucial compound for glycosylation process, thus its inhibition with 2-DG limits also sufficient viral glycosylation. I believe that we have stated our case well based on the published literature.

Minor issues 

  1. In Table 1., the lines are narrow and difficult to read, so the authors should draw a dotted line for each virus species. 
  • Table 1 has been corrected according to the Reviewer’s suggestion.

  1. For HTLV1, GLUT1 is an infection receptor, does 2-DG affect the infection of this virus more specifically? 
  • It is possible that 2-DG could bind competitively with GLUT1, inhibiting the attachment of HTLV-1 to lymphocytes. However, there is no literature that investigates 2-DG as an inhibitor of HTLV-1 infection. Future studies are needed in order to determine whether 2-DG could be a viable strategy to combat HTLV-1 infection. We have included a comment in the manuscript.

  1. In Figure 3, it would be easier to understand if the metabolic pathway of WP1122 is described. 
  • We assume that the Reviewer referred to Figure 2, where WP1122 is shown. The Figure has been corrected, and the metabolic processing of WP1122 has been included in the description.

Reviewer 2 Report

This review shows the potential of sugars, especially 2-Deoxyglucose, 2-deoxymannose and acetyl derivatives, as potential antivirals. The review is very interesting and the organization of the manuscript is excellent. The structure covers all aspects related to the pharmacological potential of these molecules, a complete analysis of the compounds is made up to their clinical phase and their antiviral activity on different viral targets. The only thing I would suggest, if the authors consider it, is to mention in a general way, the synthesis process of the new acetylated molecules. I have no objection to its being accepted in its original form.

Author Response

This review shows the potential of sugars, especially 2-Deoxyglucose, 2-deoxymannose and acetyl derivatives, as potential antivirals. The review is very interesting and the organization of the manuscript is excellent. The structure covers all aspects related to the pharmacological potential of these molecules, a complete analysis of the compounds is made up to their clinical phase and their antiviral activity on different viral targets. The only thing I would suggest, if the authors consider it, is to mention in a general way, the synthesis process of the new acetylated molecules. I have no objection to its being accepted in its original form.

  • Thank you for the positive comment and appreciation of our efforts. We corrected our manuscript according to Your suggestions. Please, find below our responses to Your comment.

The only thing I would suggest, if the authors consider it, is to mention in a general way, the synthesis process of the new acetylated molecules.

  • According to the Reviewer’s suggestion, the following statement has been added in the manuscript: “Compound WP1122 was prepared from commercially available 3,4,6-tri-O-acetyl-D-glucal in two steps synthesis. At the first step 3,4,6-tri-O-acetyl-D-glucal was selectively deacetylated to 3,6-di-O-acetyl-D-glucal, which in the next step was treated with water solution of hydrobromic acid to give WP1122 as the final product.”

Reviewer 3 Report

Pajak et coll. have reviewed  the antiviral effects of 2-DG against SARS-CoV-2  and other viruses, showing also a potential antiviral effect of a newly synthetized 2-DG analogue. The review is well organized and focused. While in cancer there are several evidences of the role of glycolysis in cell transformation for viral infection it has to be confirmed that glycolysis could be a suitable target in therapy. Therefore this is an open and recent field to be investigated. Some observations to improve the quality of the paper can be suggested:

1)Table 1. I suggest the authors to look in the literature about HTLV-1 infection and see whether it is worthwhile to add it in the list of viruses in Table 1. It has been shown that one possible way of HTLV-1  to infect the cells is involving the glucose transporter GLUT-1.

2) The authors should be more critical and precise in describing  the specificity of 2-DG versus viral infected and non-infected cells. As matter of fact  for example that in Herpes infection the inhibition of glucose metabolism could be detrimental rather than useful depending on the phase of infection.

3) The authors should be more clear regarding the mechanism of action of WP1122 in respect to 2-DG. Its better activity  is  owe to its  acetylation  favouring entrance into the cells, its pharmacokinetic  or others. The fact that it is a pro drug makes it  less susceptible to metabolism? 2-DG accumulates within the cells and inhibit HK?

4) The clinical data in SARS-Cov-2 are still ongoing in phase 1, therefore it is very difficult to design a general scenario regarding toxicity  of 2-DG,owe for example to  increase of glucose level during therapy. In conclusion  I would like to know whether,   based on their in vitro experience, the authors think  that it is conceivable that some virus infection could benefit of treatment while other not and what could be the hypothetical reason.  

Author Response

Pajak et coll. have reviewed the antiviral effects of 2-DG against SARS-CoV-2 and other viruses, showing also a potential antiviral effect of a newly synthetized 2-DG analogue. The review is well organized and focused. While in cancer there are several evidences of the role of glycolysis in cell transformation for viral infection it has to be confirmed that glycolysis could be a suitable target in therapy. Therefore this is an open and recent field to be investigated.

  • Thank you for the positive comment and appreciation of our efforts. We corrected our manuscript according to Your suggestions. Please, find below our responses to Your comments.

Some observations to improve the quality of the paper can be suggested:

1) Table 1. I suggest the authors to look in the literature about HTLV-1 infection and see whether it is worthwhile to add it in the list of viruses in Table 1. It has been shown that one possible way of HTLV-1 to infect the cells is involving the glucose transporter GLUT-1.

- The HTLV virus has been added in the Table 1 and appropriate comment in the manuscript.

2) The authors should be more critical and precise in describing the specificity of 2-DG versus viral infected and non-infected cells. As matter of fact for example that in Herpes infection the inhibition of glucose metabolism could be detrimental rather than useful depending on the phase of infection.

- Herpes Simplex virus (HSV) causes encephalitis when treated with 2-DG, likely through infection through the trigeminal ganglion. This process is possibly caused by 2-DG inhibiting the protective effect of T cells in the trigeminal ganglion during infection (Berber et al., J Immunol.  2021, 207(7), 1824-1835). We described the beneficial and detrimental effects for HSV in the text. We are aware that 2-DG action could be virus-dependent and can vary even within the same family of viruses as well as on the host cell type. We stated this also in the text.

3) The authors should be more clear regarding the mechanism of action of WP1122 in respect to 2-DG. Its better activity is owe to its acetylation favouring entrance into the cells, its pharmacokinetic or others. The fact that it is a pro drug makes it less susceptible to metabolism? 2-DG accumulates within the cells and inhibit HK?

- WP1122 is a 2-DG prodrug. As described in the text (line 563), “WP1122 enters the cells via passive diffusion rather than relying upon specific glucose transporters. Inside the cells, WP1122 undergoes deacetylation by esterases releasing active 2-DG molecules (Figure 2). Further, 2-DG undergoes phosphorylation at the C-6-hydroxyl group, and it is trapped inside the cells. 6-phosho-2-DG competitively inhibits HK, blocking phosphorylation of glucose and thereby inhibiting the glycolytic pathway.” We could add that the 2-DG produced from WP1122 which is already intracellular feeds into the pathways described in Figure 1, thus bypassing competition for glucose transporters as exogenous 2-DG does. Thus, it owes its superior activity as compared to 2-DG by enhanced uptake into cells, no competition for a transporter and reduced metabolism when extracellular. 

4) The clinical data in SARS-Cov-2 are still ongoing in phase 1, therefore it is very difficult to design a general scenario regarding toxicity of 2-DG, owe for example to increase of glucose level during therapy. In conclusion I would like to know whether, based on their in vitro experience, the authors think that it is conceivable that some virus infection could benefit of treatment while other not and what could be the hypothetical reason. 

- Thank you very much for this important comment. As correctly indicated here it is possible, if not likely, that some hospitalized patients receiving i.v. fluids may also receive glucose at 5%, 10% or even higher (not limited to COVID-19 patients). This can be especially the case for patients that are intubated and unable to drink and eat. However, our primary goal is to reduce hospitalization rates and improve recovery through early treatment of patients that are not receiving i.v. fluid therapy yet, especially in outpatient group of COVID-19 infected population.

With regards to the second part of the question, our group is not aware of any specific negative impact of this type of therapy against other viral infections. The only exception could be related to Herpes infection and that is being addressed in the other section of the paper. In general, it is certainly possible that patients receiving glucose i.v. therapies may not benefit from 2-DG but we can only speculate at this stage. Nevertheless, we have included short description in this revised version of our paper to address this important issue.